# Boundary-Aware Periodicity-based Sparsification Strategy for Ultra-Long Time Series Forecasting

## ABSTRACT

In various domains such as transportation, resource management, and weather forecasting, there is an urgent need for methods that can provide predictions over a sufficiently long time horizon to encompass the period required for decision-making and implementation. Compared to traditional time series forecasting, ultra-long time series forecasting requires enhancing the model's ability to infer long time series, while maintaining inference costs within an acceptable range. To address this challenge, we propose the **B**oundary-**A**ware **P**eriodicity-based sparsification strategy for **U**ltra-**L**ong time series forecasting (**BAP-UL**). The periodicity-based sparsification strategy is a general lightweight data sparsification framework that captures periodic features in time series and reorganizes inputs and outputs into shorter sub-sequences for model prediction. The boundary-aware method, combined with the bounded nature of time series, improves the model's capability to predict extreme peaks and irregular time series by adjusting the prediction results. We conducted extensive experiments on benchmark datasets, and the BAP-UL model achieved nearly **90%** state-of-the-art results under various experimental conditions. Moreover, the data sparsification method based on the periodicity, proposed in this paper, exhibits broad applicability. It enhances the upper limit of sequence length for mainstream time series forecasting models and achieves the state-of-the-art prediction results.

## CCS CONCEPTS

• **Computing methodologies → Temporal reasoning**.

## KEYWORDS

Ultra-Long Time Series, Boundary, Periodic, Sparsification Strategy

## 1 INTRODUCTION

In multimedia data processing tasks, time series forecasting models must analyze multiple variables from different dimensions[22]. For example, in autonomous driving field[1], the models need to simultaneously analyze variables including time, vehicle speed, satellite lane maps, ground height, and sensor images. This imposes significant challenges on the model's capacity to manage multiple variables. In this scenario, increasing the length of the predicted time series would result in a significant increase in computational

*ACM MM, 2024, Melbourne, Australia*
© 2024 Copyright held by the owner/author(s). Publication rights licensed to ACM.
ACM ISBN 978-x-xxxx-xxxx-x/YY/MM
https://doi.org/10.1145/nnnnnnn.nnnnnnn

**Unpublished working draft. Not for distribution.**

costs. This becomes a barrier to extending the model's predicted sequence length.

In diverse fields such as transportation, resource management, and weather forecasting, the prediction horizon offered by conventional models in long time series forecasting (LTSF) frequently falls short of encompassing the duration necessary for decision-making and implementation to yield results, owing to the involvement of numerous factors and intricate adjustment processes[2]. Among the methods capable of LTSF, the Transformer model relies on the data sparsification strategy, which may lead to a decline in the model's robustness. In contrast, linear models can directly infer longer sequences, but the current prediction length of 720 points is already at the performance boundary[19, 30].

On the flip side, while current state-of-the-art (SOTA) models are capable of forecasting up to 720 time points into the future, their full potential in LTSF is often hindered by the substantial computational demands associated with handling multidimensional and multivariate data. For instance, in the task of predicting daily advertising costs [22], fine-grained data needs to be aggregated on a daily basis to achieve cost predictions for each day over the next 180 days. Clearly, predictions at a finer time granularity are more helpful for devising advertising placement plans. Therefore, enhancing the model's ability to predict longer sequences while maintaining high performance in handling large-scale datasets with multiple variables and dimensions is a challenging yet crucial research area in time series forecasting.

To address this challenge, we propose the **B**oundary-**A**ware **P**eriodicity-based sparsification strategy for **U**ltra-**L**ong time series forecasting (**BAP-UL**). The BAP-UL model exhibits a substantial enhancement in prediction performance on ultra-long time series, multi-variable datasets, significantly surpassing existing benchmark models. As depicted in Figure 1, it presents the **M**ean **S**quared **E**rror (**MSE**) for each model on three benchmark datasets with a higher number of variables, where the input length is set at 720 and the prediction length extends to 2880. The periodicity approach is a lightweight and generalizable framework that captures the periodic features in sequences using the Fast Fourier Transform (FFT). It reconstructs the input-output sequences of the model into multiple shorter sub-sequences with non-overlapping time points, based on the captured periodic features. The inferred sub-sequences are then combined using the periodic features to recover the complete prediction sequence, enabling the processing of the entire ultra-long time series. The boundary-aware method, leveraging the bounded nature of time series, introduces maximum and minimum boundary prediction to enhance the model's capability in predicting extreme peaks and irregular time series.

When tested on publicly available benchmark datasets, the BAP-UL model consistently achieved optimal results of nearly **90%** under different experimental conditions. Furthermore, the data sparsification method proposed in this paper exhibits broad generality.

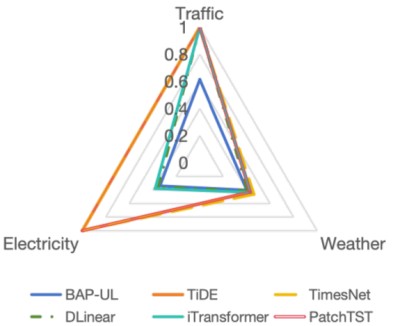

**Figure 1: Comparison of MSE between the BAP-UL model and baseline models. All models have an input length of 720 and a prediction length of 2880.**

When combined with mainstream time series forecasting models, it significantly increases the length of input-output sequences that can be handled, leading to superior SOTA results in an effective manner. In summary, our **contributions** are as follows:

(1) We propose the BAP-UL model, which can perform **U**ltra-**L**ong **T**ime **S**eries **F**orecasting (**ULTSF**) tasks with an input of 720 and an output of 2880. It can handle large-scale datasets with multiple dimensions and variables.

(2) We present the periodicity approach, a universal lightweight data sparsification framework that enhances the prediction capability of mainstream forecasting models.

(3) Extensive experiments were conducted on three benchmark datasets containing a larger number of variables. The results achieved nearly 90% of the state-of-the-art performance across various experimental configurations.

(4) On the three benchmark datasets, we found that leveraging periodicity-based sparsity strategy can enhance the prediction capabilities of baseline models on ultra-long time series.

## 2 RELATED WORK

### 2.1 Advancements in Model Architecture

With the development of deep learning, significant successes have been achieved in various fields, including Natural Language Processing (NLP) [9, 25], Computer Vision (CV) [10, 13, 37], and Time Series Forecasting (TSF) [18, 21, 36].

Early TSF models were based on Recurrent Neural Networks (RNNs) [3, 20, 24, 26, 29, 33], and the two classic algorithms proposed by researchers, LSTM [11] and GRU [7], are still widely used to this day [5, 15, 17]. However, due to the issue of gradient vanishing or exploding in handling long sequences, the performance of representative methods based on RNN models, such as DeepAR [27], is not satisfactory for LTSF. Based on Convolutional Neural Networks (CNNs) models, the gradient propagation during training is more stable, leading to the development of numerous models with strong prediction capabilities, such as LSTnet [14], TCN [4, 28], etc. Among them, MICN [31] introduces a mechanism for temporal decomposition, and TimesNet [34] achieves the SOTA results by combining the time series forecasting task with 2D image.

The Informer model [38] pioneered the task of LTSF by increasing the input length to 96 and the output length to 720, achieving substantial success. This has stimulated the development of many variant models based on the Transformer architecture, such as Reformer [12], Autoformer [35], and FEDformer [39]. These new models set new SOTA standards for LTSF results. It was not until the emergence of the DLinear [36], which achieved better results in LTSF experiments using a simple linear structure, that the applicability of the Transformer architecture to LTSF was questioned. Subsequent researchers proposed methods such as PatchTST [21], which segments subsequences, and iTransformer [18], which flips the time and feature dimensions, both of which once again demonstrated the superiority of the Transformer architecture in LTSF. There are also linear-based models for LTSF tasks, such as N-BEATS [23] and NHiTS [6], which are based on Multi-Layer Perceptron (MLP) architectures. Among them, the TiDE model [8] proposed by Google in 2023 achieved the SOTA performance at that time by utilizing MLP with residual networks.

### 2.2 Advancements in Time Series Length

In multimedia data processing tasks, time is a commonly used fundamental dimension for various prediction tasks, often employed to support short-term scheduling [1, 2] and long-term planning [22] . The task of time series forecasting has traditionally focused on short-term forecasting, with prediction lengths ranging from 1 to 48. Until the proposal of Informer [38], the prediction length was improved to a range between 96 and 720 by leveraging the long sequence processing capability of Transformer. In the following three years, the upper limit for forecast lengths remained at 720.

Although the Transformer architecture has the ability to handle long sequences, its quadratic computation cost leads to a significant increase in computational and memory resources as the sequence length grows. In order to achieve LTSF, many Transformer-based models adopt the data sparsification strategy. For example, Log-Trans [16] utilizes logarithmic sparse attention, and Informer [38] selects a subset of queries based on Kullback-Leibler divergence during attention computation. However, [32] has indicated that these models experience a significant drop in prediction performance when only the input sequence length is increased.

Currently, prediction models based on linear architectures have gained significant attention. These models exhibit excellent performance in LTSF tasks. However, as the sequence length increases, the computational cost of these models also grows rapidly. Existing models in this category are limited to sequence lengths up to 720 for prediction [19, 30]. One potential approach to tackle this problem is by employing effective sub-sequence serialization methods that keep the sequence lengths within the model's operational limits. Additionally, these methods should be able to restore the predicted sub-sequences to the desired prediction length.

## 3 METHOD

Our BAP-UL model consists of two main components: the **periodicity-based sparsification strategy** and the **boundary-aware method**. The periodicity-based sparsification strategy forms the foundation

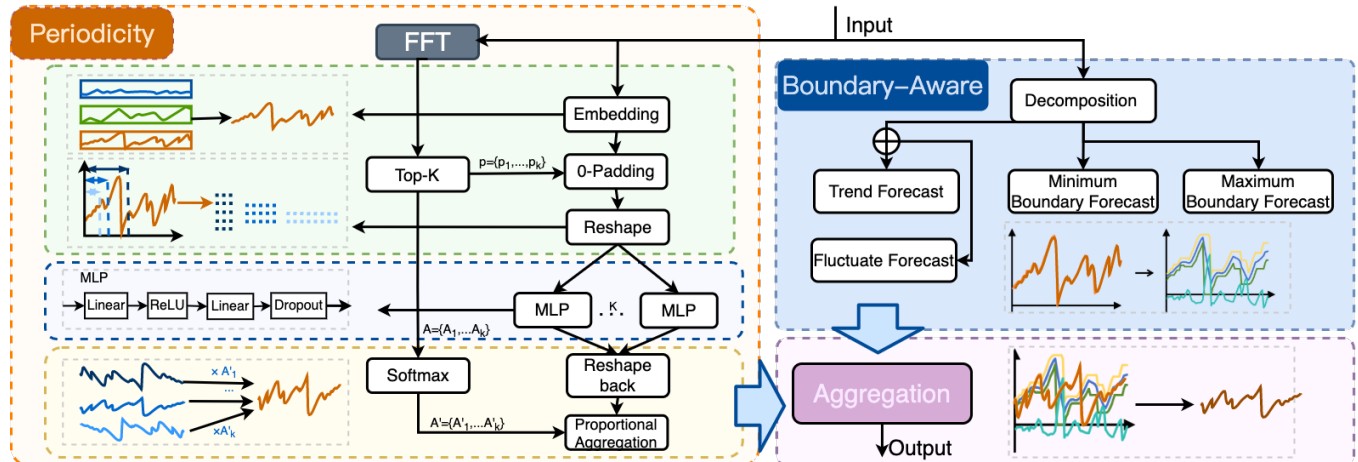

**Figure 2: BAP-UL model overview. The Periodicity part is responsible for capturing periodic features in the sequence and reorganizing the input and output sequences into short sub-sequences for prediction. The Boundary-Aware part is responsible for combining the boundedness of the time series to enhance the model's ability to predict extreme peaks and irregular sequences. The Aggregation part is responsible for aggregating the prediction results from the Periodicity part and the Boundary-Aware part to obtain the final result of the BAP-UL model.**

for ULTSF. The boundary-aware method enhances the model's ability to predict extreme peaks and irregular time series. The overall architecture of the BAP-UL model is detailed in Figure 2.

### 3.1 Periodicity-based Sparsification Strategy

In this subsection, we will provide a detailed explanation of the periodicity-based sparsification strategy proposed in this paper. It mainly consists of four components, namely Periodic Feature Extraction, Periodic Reorganization, Sub-sequence Prediction, and Prediction Result Aggregation.

*3.1.1 **Periodic Feature Extraction**.* First, the periodicity-based sparsification strategy utilizes the Fast Fourier Transform (FFT) to extract the periodic features of the entire dataset. In this paper, we utilize the periodic features extracted by FFT to reorganize the input sequence into multiple shorter sub-sequences with no overlapping time points, separated by intervals equal to the period, thereby reducing the sequence length.

The model takes a time series segment $X \in \mathbb{R}^{L \times M}$ as input, where $L$ is the input length, and there are $M$ variables in total. By using the FFT operation, we can obtain the amplitudes of various frequencies in the dataset. Since there is no periodicity when the frequency is 0, and the frequency cannot be greater than $L$, it is sufficient to calculate the average amplitude $A$ corresponding to frequencies ranging from 0 to $L$. Then, the $K$ highest amplitude values $\{A_1, ..., A_K\}$ and their corresponding frequencies $\{f_1, ..., f_K\}$ are selected from this range. The $i$-th frequency indicates its occurrence $f_i$ times in the sequence, and the corresponding period length $p_i$ is the total length of the sequence $L$ divided by the frequency $f_i$, expressed as $p_i = L/f_i$. Using the above equation, the $K$ frequencies can be converted into periods within a backtrack window of length $L$, denoted as $\{p_1, ..., p_K\}$. The amplitudes $A$ and periods $p$ constitute the periodic features of the dataset.

*3.1.2 **Periodic Reorganization**.* To further enhance the model's ability to handle large-scale datasets with a large number of variables, we map the $M$ variables in the original dataset to a low-dimensional space in the Embedding layer, reducing the number of parameters in the model. Then, the data within the backtracking window is reorganized based on periodicity. For the $i$-th periodic feature $p_i$, the sequence will be reorganized into $p_i$ sub-sequences. At this point, the model only needs to predict each of the $p_i$ sub-sequences separately. The principle of periodic reorganization is illustrated in Figure 3.

In cases where there's no discernible periodicity in the input data, it's regarded as having an infinitely large period, effectively equating to a frequency of 0. In such instances, rather than allowing the entire input sequence to degrade the model's performance, these situations will be filtered out. Conversely, if the input data exhibits periodicity with a unit period of 1, corresponding to a frequency of $L$, the model's input sequence will be segmented into $L$ sub-sequences, each consisting of a single time point. Consequently, the length of the model's input sequence will be reduced to 1, minimizing the computational complexity of the model.

The periodic reorganization task pads the length of $X$ to be an integer multiple of the period $p_i$, with placeholders set to 0 in this paper. After padding, the data length becomes $\hat{L}_i$. Then, the data is sampled at intervals of the period length $p_i$, forming sub-sequences by grouping points at the same relative positions within different cycles. After reorganization, $\hat{X}_i$ takes the form $[p_i, \hat{L}_i/p_i]$, where $0 \leq i \leq K$. The output of cyclic reorganization is $\hat{X}_i$, where the data in the $n$-th row $\hat{X}_i^n$, with $0 \leq n \leq p_i$, represents the time sub-sequence composed of the $n$-th point within each periodic under the $i$-th cycle feature $p_i$. The calculation formula for periodic

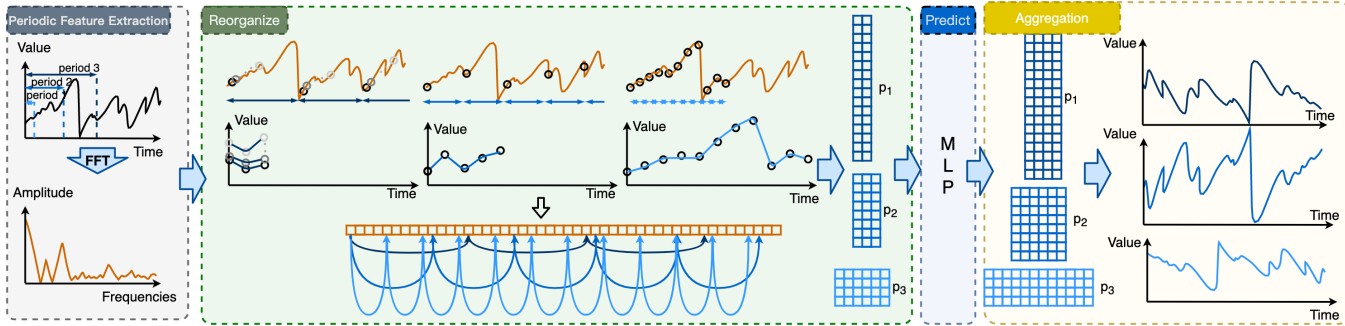

Figure 3: The illustration demonstrates time series forecasting using a Periodicity-based Sparsification Strategy.

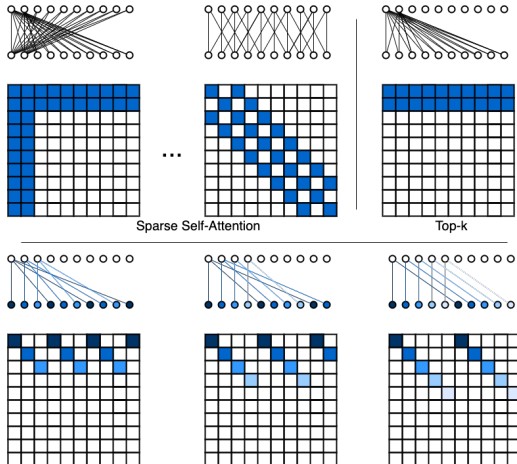

Figure 4: Comparison of Periodic Data Sparsification Methods and Traditional Time Series Sparsification Methods

reorganization is as follows:

$$\hat{X}_i = Reshape(Padding(Embedding(X), \left\lceil \frac{L}{p_i} \right\rceil \cdot p_i), p_i), 0 \le i \le K \tag{1}$$

Unlike traditional strategies such as selecting only the top-k or attention-based sparsification, the sub-sequences obtained through periodic reorganization are ordered, and each sub-sequence has a fixed length. This type of sparsification strategy can better preserve the features of the original data while keeping the computational complexity at an appropriate level.

*3.1.3* ***Sub-sequence Prediction.*** The input for the subsequence prediction task is the reorganized subsequence $\hat{X}_i^n$ with a length of $\hat{L}_i / p_i$, and the output is a predicted subsequence with a length of $\hat{P}_i = \left\lceil \frac{P}{p_i} \right\rceil \cdot p_i$, here $P$ is the prediction length. $\hat{X}_i^n$ is passed through an MLP structure for prediction separately, resulting in predicted subsequences $\dot{X}_i, 0 \le i \le K$. $\dot{X}_i$ has a size of $[p_i, \hat{P}/p_i]$. Then, the predicted subsequences $\dot{X}_i$ are reorganized into a predicted sequence with a length of $\hat{P}_i$, and the first P predicted values are extracted as the final predicted sequence $\ddot{X}_i$. The calculation formula

for subsequence prediction is as follows:

$$\dot{X}_i^n = MLP(\hat{X}_i^n, \left\lceil \frac{P}{p_i} \right\rceil \cdot p_i), \quad 0 \le n \le p_i \tag{2}$$

$$\ddot{X}_i = Reshape(\dot{X}_i), \quad 0 \le i \le K \tag{3}$$

When the prediction length $P$ is twice the input length $L$, the principle of sub-sequence prediction is illustrated in Figure 3.

*3.1.4* ***Prediction Result Aggregation.*** Using the $Softmax$ function to normalize the periodic feature $A$, obtaining K proportion coefficients for the K predicted results $\ddot{X}_i$. Then, aggregating the predicted results of each sub-sequence by multiplying them by their respective proportion coefficients. After mapping the variables back to M, the final prediction result $X_{PF}$ is obtained. The calculation formula for periodic forecasting (PF) method is as follows:

$$X_{PF} = projection(Softmax(A) \times \ddot{X}) \tag{4}$$

## 3.2 Boundary-Aware Method

The boundary-aware method decomposes the time series into trend fluctuations, maximum boundaries, and minimum boundaries. It predicts future time series transformations from three perspectives, forming a prediction system sensitive to upper and lower boundaries, overall trends, and fluctuations. In the boundary-aware method, for input data $X \in \mathbb{R}^{L \times M}$, when the pooling window size is $n$, the sequence is first padded to a length of $L + n$ to compute the moving average. The $Repeat$ operation is then used to copy the first and last elements of the sequence $n/2$ times each, resulting in a sequence of length $L + n$.

$$X_{L+n} = Repeat(X, \frac{n}{2}) \tag{5}$$

Using the $AvgPool1d$ function with a pooling window size of $n$ and a stride of $s$, compute the moving average to obtain the trend component. Then, subtract the trend component from the input data to obtain the fluctuation component. Predict each component separately and add the predicted results together to obtain the prediction result of the trend-fluctuation module $X_{TFF}$. The calculation formula is as follows:

$$X_{Trend} = AvgPool1d(X_{L+n}, n, s) \tag{6}$$

$$X_{TFF} = Linear(X_{Trend}) + Linear(X - X_{Trend}) \tag{7}$$

The maximum boundary prediction module employs the $MaxPool1d$ function to compute the moving maximum of the sequence $X$ with

**Algorithm 1:** BAP-UL Algorithm Structure

---

$A, p \leftarrow \text{PF}(X)$ ;
$X_E \leftarrow \text{Embedding}(X, C)$ ;
**for** $i = 0 \text{ to } K$ **do**
    $\hat{X}_i \leftarrow \text{Reshape}(\text{Padding}(X_E, p_i), p_i)$ ;
    **for** $n = 0 \text{ to } p_i$ **do**
        $\tilde{X}_i^n \leftarrow \text{MLP}(\hat{X}_i^n)$ ;
    **end**
**end**
$X_{PF} \leftarrow \text{projection}(\text{Softmax}(A) \times \tilde{X})$ ;
$X_{TFF}, X_{MaxF}, X_{MinF} = LinearF(Decomp(Repeat(X, \frac{n}{2})))$ ;
$Y = Linear(Stack(X_{TrendF}, X_{FluF}, X_{MaxF}, X_{MinF}, X_{PF}))$ ;

---

a pooling window of $n$ and a stride of $s$, yielding the maximum boundary. Subsequently, the maximum boundary is passed through the Linear function for prediction, resulting in the prediction result of the maximum boundary module $X_{MaxF}$. The calculation formula for the maximum boundary prediction module is as follows:

$$X_{MaxF} = Linear(MaxPool1d(X_{L+n}, n, s)) \tag{8}$$

The minimum pooling utilizes the *MinPool* function, with a pooling window of $n$ and a stride of $s$, to compute the moving minimum of the sequence $X$. The moving minimum $X_{Min}[i, j]$ represents the minimum value for the $j$-th variable from the $i$-th time step to the $i+n$-th time step. Finally, the minimum boundary is passed through the Linear function for prediction, resulting in the prediction result of the minimum boundary module $XMinF$. The calculation formula for the minimum boundary prediction module is as follows:

$$MinPool(X[i, j], n, s) = \min(X_{L+n}[i \cdot s : i \cdot s + n, j]) \tag{9}$$

$$X_{MinF} = Linear(MinPool(X_{L+n}, n, s)) \tag{10}$$

Finally, the model combines the results of periodic prediction with those of boundary-aware prediction to obtain the final prediction result $Y$. The computational formula for the model is as follows:

$$Y = Linear(Stack(X_{TFF}, X_{MaxF}, X_{MinF}, X_{PF})) \tag{11}$$

The method of decomposing and then predicting the time series described above can be summarized in the following formula:

$$X_{TFF}, X_{MaxF}, X_{MinF} = LinearF(Decomp(Repeat(X, \frac{n}{2}))) \tag{12}$$

The algorithmic structure of the model is illustrated in the 1.

## 4 EXPERIMENT

In this paper, we have selected six well-acknowledged models with different structures as our baseline models. These models have been widely recognized in the literature for their superior performance in time series forecasting tasks. Given that our data sparsification strategy is primarily intended to tackle the problem of high computational costs associated with managing ultra-long sequences and a multitude of variables in forecasting tasks, the experimental datasets chosen for this paper include Traffic, Electricity, and Weather datasets, all of which entail a significant number of variables [35]. When the input length is set to 720, the detailed information about the data partitioning for the three benchmark datasets is presented in Table 1.

**Table 1: Details of the three benchmark datasets.**

| Dataset | VaS | (Train,Val,Test) | Fre | Span |
|---------|-----|------------------|-----|------|
| Traffic | 862 | (10526,3510,3508) | 1 Hour | 120Days |
| Electricity | 321 | (31617,10540,10539) | 1 Hour | 120Days |
| Weather | 21 | (15782,5262,5260) | 10 MIN | 20Days |

### 4.1 Experimental Results

Table 2 presents the complete results for the ultra-long time series forecasting task. The experimental setup is the same for each model, with an input length of 720 and prediction lengths of 1440, 2160, 2880. "-" indicates that the model encountered an out-of-memory (OOM) error in this experimental environment. The number in brackets below the dataset name represents the number of variables in the dataset. Lower MSE and MAE values indicate better prediction performance for the model. The best results are bolded, and the second-best results are underlined.

It can be observed that our BAP-UL model demonstrates a significant advantage in the capability of ultra-long time series forecasting, achieving optimal prediction performance in nearly 90% of the experimental settings. Moreover, it exhibits a distinct advantage in terms of memory complexity. This is evident from the occurrence of varying degrees of memory overflow in the benchmark models included in the comparison, especially the TiDE, TimesNet, and PatchTST models, which are characterized by higher complexity. In contrast, our BAP-UL model did not encounter such issues during training.

Further analysis of the memory overflow problem reveals that some models experienced memory overflow when the prediction length was set to 1440 on the traffic dataset. However, they were able to execute properly when the prediction length was increased to 2160. This can be attributed to the fact that a smaller number of samples are generated when the prediction length is 2160, resulting in lower memory consumption.

In the evaluation experiments on the Weather dataset, we observed that existing mainstream models are capable of handling ultra-long time series forecasting tasks on datasets with a smaller number of variables.

When the number of variables increased to 862, all benchmark models experienced OOM errors when the prediction length was 1440 or 2880, demonstrating the superior multivariate processing capability of our model. The lower half of Figure 5 shows a comparison of the prediction results between the benchmark models that did not encounter OOM and the BAP-UL model on the Electricity and Traffic datasets when the number of variables was large. The BAP-UL model is capable of predicting subtle peak variations within different periods, exhibiting superior forecasting performance.

We also found that none of the models were able to accurately predict extreme peak values, which hold significant implications in practical applications. For instance, in the field of transportation, when certain dates experience excessively high traffic flow that requires diversion, a forecast that indicates a regular peak value could miss the opportunity for traffic management, thus failing to prevent severe congestion. Therefore, finding ways to better

**Table 2: Prediction results of the multivariate Ultra-long time series forecasting task on three benchmark datasets.**

| Models | | BAP-UL | | TiDE | | TimesNet | | DLinear | | iTransformer | | PatchTST | |
|---|---|---|---|---|---|---|---|---|---|---|---|---|---|---|
| Metric | | MSE | MAE | MSE | MAE | MSE | MAE | MSE | MAE | MSE | MAE | MSE | MAE |
| Weather (21) | 1440 | **0.376** | 0.398 | 0.389 | 0.389 | 0.395 | 0.393 | 0.378 | 0.397 | 0.398 | 0.396 | 0.401 | 0.394 |
| | 2160 | **0.388** | 0.410 | 0.416 | 0.408 | 0.440 | 0.427 | 0.389 | 0.420 | 0.426 | 0.414 | 0.407 | 0.403 |
| | 2880 | **0.402** | 0.428 | 0.432 | 0.421 | 0.463 | 0.433 | 0.409 | 0.439 | 0.436 | 0.426 | 0.431 | 0.423 |
| Electricity (321) | 1440 | **0.263** | 0.366 | - | - | 0.293 | 0.367 | 0.275 | 0.357 | 0.286 | 0.356 | - | - |
| | 2160 | **0.282** | 0.374 | - | - | - | - | 0.318 | 0.387 | 0.351 | 0.405 | - | - |
| | 2880 | **0.339** | 0.414 | - | - | - | - | 0.353 | 0.412 | 0.380 | 0.424 | - | - |
| Traffic (862) | 1440 | **0.557** | 0.366 | - | - | - | - | - | - | - | - | - | - |
| | 2160 | 0.570 | 0.380 | - | - | - | - | 0.578 | 0.367 | **0.527** | 0.341 | - | - |
| | 2880 | **0.615** | 0.390 | - | - | - | - | - | - | - | - | - | - |

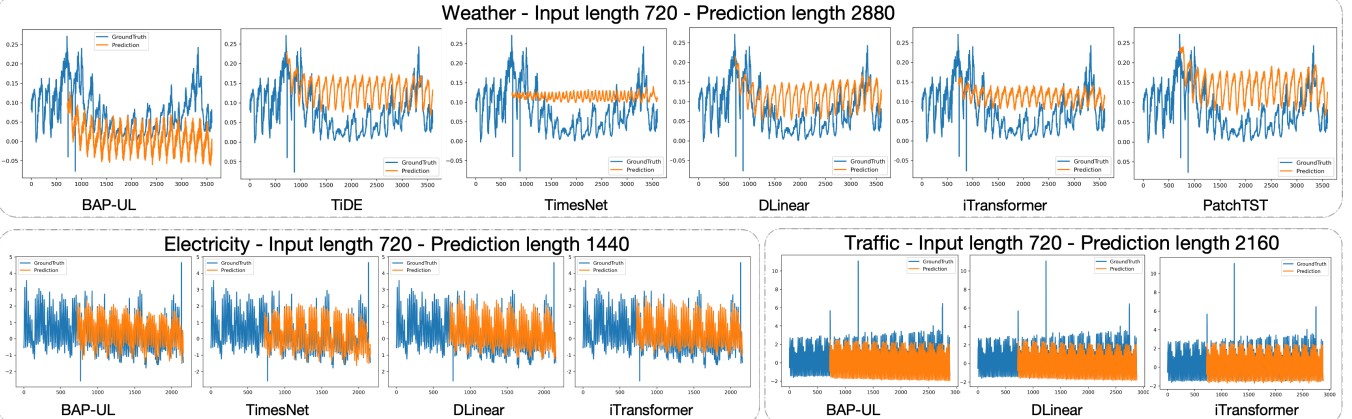

**Figure 5: Visualization of the prediction results of all models on the benchmark datasets.**

predict datasets with such extreme peak values represents a highly valuable direction for future research.

## 4.2 Generalization Study

We further investigated the generality of the periodicity-based sparsification strategy by combining it with benchmark models under different experimental settings. Three models were chosen based on their successful execution in certain experimental settings. Comparative experiments were conducted by embedding these models into the periodicity module, and the changes in prediction performance were evaluated using MSE as the evaluation metric.

Table 3 presents the results of the experiments on the generalization of the periodicity framework. In this table, "*" indicates the model that incorporates the benchmark model with the periodicity-based sparsification strategy. The experimental setup is the same for each model, with an input length of 720 and prediction lengths of 1440, 2160, 2880. "-" indicates that the model encountered an OOM error in this experimental environment. Lower MSE values indicate better prediction performance for the model. Blue numbers indicate an improvement in prediction ability compared to the original model. The best prediction performance is bolded. When the outer layers of different model architectures are embedded with the

periodicity-based method proposed in this paper, the LTSF capability of each benchmark model is significantly improved. In 77% of the experimental settings, a reduction in MSE loss is observed. Additionally, the optimal results in 5 experimental settings have been updated. For example, on the Weather dataset, SOTA is reduced from 0.376 to 0.365 after incorporating the periodicity framework.

Our proposed periodicity-based method is a lightweight framework. It's evident from the Traffic dataset that integrating the periodicity framework does not exacerbate memory overflow issues. Instead, it aids in reducing the memory consumption of the models. As shown in the table, the TimesNet, DLinear, iTransformer, and Transformer models all successfully predict sequences of length 2880 without encountering memory overflow.

We also observed that the iTransformer, which integrates reversed variables and temporal dimensions, does not demonstrate significant advantages in terms of prediction capability compared to the traditional iTransformer. In one-third of the experimental settings, the Transformer model outperforms the iTransformer.

## 4.3 Experimental Analysis

*4.3.1 Ablation study.* The final prediction results of the BAP-UL model are derived from four methods: the periodicity framework, denoted as PF, and three boundary-aware prediction methods (BA),

**Table 3: Generalization experiment results of the periodization framework.**

| Models | | BAP-UL | TimesNet* | TimesNet | DLinear* | DLinear | iTransformer* | Transformer* | iTransformer |
|---|---|---|---|---|---|---|---|---|---|
| Weather | 1440 | 0.376 | 0.372 | 0.395 | 0.366 | 0.378 | **0.365** | 0.370 | 0.398 |
| | 2160 | 0.388 | 0.393 | 0.440 | 0.389 | 0.389 | **0.381** | 0.397 | 0.426 |
| | 2880 | 0.402 | **0.382** | 0.463 | 0.407 | 0.409 | 0.395 | 0.391 | 0.436 |
| Electricity | 1440 | 0.265 | **0.258** | 0.293 | 0.275 | 0.275 | 0.280 | 0.262 | 0.286 |
| | 2160 | **0.282** | - | - | 0.294 | 0.318 | 0.327 | 0.322 | 0.351 |
| | 2880 | 0.339 | - | - | **0.305** | 0.353 | 0.355 | 0.367 | 0.380 |
| Traffic | 1440 | **0.557** | 0.580 | - | 0.624 | - | 0.575 | - | - |
| | 2160 | 0.570 | 0.577 | - | 0.585 | 0.578 | 0.590 | 0.660 | **0.527** |
| | 2880 | **0.615** | 0.647 | - | 0.649 | - | 0.640 | 0.667 | - |

namely Trend Fluctuation Forecasting (TFF), Maximum Boundary Forecasting (MaxF), and Minimum Boundary Forecasting (MinF). In order to evaluate the effectiveness of our method, we conducted extensive ablation experiments and the complete results can be found in Table 4. Each model was evaluated under the same experimental settings, with an input length of 720 and prediction lengths of 1440, 2160, 2880. The "N-" prefix indicates that the corresponding sub-module (X) was excluded from the overall model structure. "-" indicates that the model encountered an OOM error in the experimental environment. Lower MSE values indicate better prediction performance of the models. The best prediction performances are highlighted in bold, and the second-best performances are underlined.

The ablation experiments indicate that when the periodicity framework PF is used alone for prediction tasks, the prediction performance is not satisfactory. The boundary-aware prediction method BA module itself also has good prediction ability, and the introduction of the BA module can effectively improve the overall prediction performance of the model. Furthermore, when examining the overall results, it can be observed that after selectively disabling a specific submodule, the prediction results exhibit a certain degree of decline compared to the BAP-UL model. However, the overall performance still remains at a suboptimal level, indicating that each submodule plays a role in correcting and refining the prediction results.

*4.3.2 Feasibility of Periodicity-based Sparsification Strategy.* To analyze the effectiveness of the proposed periodicity-based sparsification strategy in this paper, visual analysis of the overall dataset as well as the training, validation, and testing sets was conducted to examine their respective periodic characteristics, as shown in Figure 6.

In this figure, darker colors indicate higher amplitudes at corresponding frequencies. The amplitudes of frequencies across different sets exhibit a long-tail distribution, with only a very small number of frequencies having larger amplitudes. The training set, due to its large volume of data, exhibits consistent characteristics with the overall dataset. However, slight differences can be observed in the periodic features of the validation and testing sets compared to the overall dataset. This suggests that the periodic features in the testing set accurately reflect the periodic characteristics of the entire dataset, demonstrating their feasibility.

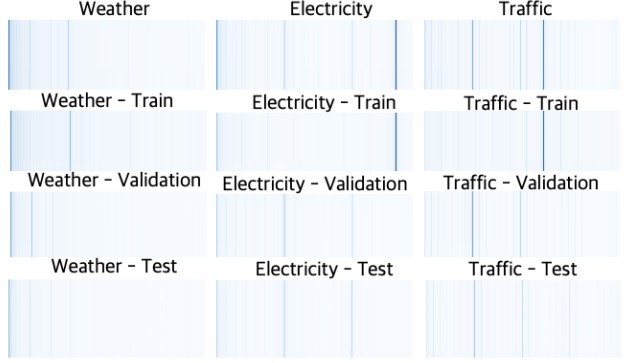

**Figure 6: Visualization of Overall and Training, Validation, and Test Set Periodic Features for the three benchmark datasets.**

*4.3.3 Model Complexity.* The BAP-UL model has two parameters: the number of periodic features, denoted as $k$, and the model dimension, denoted as $d_{model}$. The number of periodic features, $k$, corresponds to the number of stacked layers in the MLP structure within the model. The model dimension, $d_{model}$, represents the size to which the variables are compressed within the model. To compare the effects of different combinations of $k$ and $d_{model}$ on different datasets, six sets of comparative experiments were conducted. The experimental results are shown in Figure 7.

It is evident that the impact of different $k$ values on the model's prediction results is not significant. Surprisingly, lower values of $d_{model}$, indicating higher compression of variables, result in lower model loss. The model is capable of significantly compressing the number of variables to be computed for each dataset. For example, in the Traffic dataset with 862 variables, the model achieves the best performance with a $d_{model}$ value of only 128, reducing the computational complexity by 73%. Therefore, this model is highly suitable for large-scale prediction datasets with a large number of variables, such as predicting server resource usage in datasets with tens of thousands of distributed server nodes.

Table 4: Ablation Experiment Results.

| Models | | BAP-UL | | PF | | BA | | N-TFF | | N-MaxF | | N-MinF | | iTransformer | |
|---|---|---|---|---|---|---|---|---|---|---|---|---|---|---|---|
| Metric | | MSE | MAE | MSE | MAE | MSE | MAE | MSE | MAE | MSE | MAE | MSE | MAE | MSE | MAE |
| Weather | 1440 | 0.376 | 0.398 | 0.399 | 0.394 | **0.368** | 0.397 | 0.386 | 0.412 | 0.371 | 0.401 | 0.375 | 0.406 | 0.398 | 0.396 |
| | 2160 | **0.388** | 0.410 | 0.438 | 0.420 | **0.388** | 0.415 | 0.392 | 0.416 | 0.403 | 0.431 | 0.399 | 0.426 | 0.426 | 0.414 |
| | 2880 | **0.402** | 0.428 | 0.463 | 0.428 | 0.405 | 0.433 | 0.415 | 0.442 | 0.427 | 0.459 | 0.420 | 0.450 | 0.436 | 0.426 |
| Electricity | 1440 | **0.263** | 0.366 | 0.293 | 0.375 | 0.286 | 0.371 | 0.277 | 0.381 | 0.274 | 0.374 | 0.294 | 0.382 | 0.286 | 0.356 |
| | 2160 | **0.282** | 0.374 | 0.341 | 0.403 | 0.332 | 0.402 | 0.312 | 0.399 | 0.326 | 0.400 | 0.338 | 0.412 | 0.351 | 0.405 |
| | 2880 | 0.339 | 0.414 | 0.512 | 0.495 | 0.355 | 0.415 | 0.405 | 0.458 | 0.352 | 0.418 | **0.329** | 0.405 | 0.380 | 0.424 |
| Traffic | 1440 | **0.557** | 0.366 | 0.746 | 0.403 | - | - | 0.794 | 0.463 | 0.574 | 0.367 | 0.669 | 0.399 | - | - |
| | 2160 | 0.570 | 0.380 | 0.757 | 0.404 | 0.624 | 0.398 | 0.807 | 0.449 | 0.619 | 0.391 | 0.645 | 0.401 | **0.527** | 0.341 |
| | 2880 | **0.615** | 0.390 | 0.797 | 0.414 | 0.658 | 0.401 | 0.775 | 0.420 | 0.628 | 0.400 | 0.684 | 0.440 | - | - |

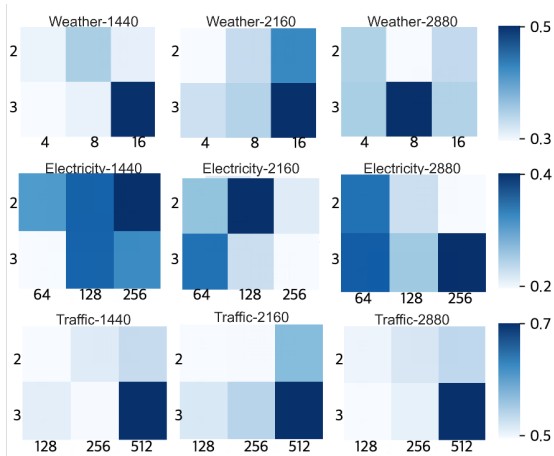

**Figure 7: Comparison of MSE values for different values of $k$ (the number of periodic features) and $d_{model}$ (the model dimension). Comparative experiments were conducted with $k$ values of 2 and 3 for all datasets. For the Weather dataset, $d_{model}$ was set to 4, 8, 16. For the Electricity dataset, $d_{model}$ was set to 64, 128, 256. For the Traffic dataset, $d_{model}$ was set to 128, 256, 512. Lighter colors in the corresponding experimental settings indicate lower MSE values, indicating better prediction performance of the models.**

## 5 CONCLUSION

In this paper, we propose the Boundary-Aware Periodicity-based sparsification strategy for Ultra-Long time series forecasting (BAP-UL), which can perform with an input length of 720 and an output length of 2880. Meanwhile, it can handle large-scale datasets with multiple dimensions and variables. Furthermore, we introduce the periodicity approach, a versatile lightweight data sparsification framework that enhances the prediction capability of mainstream forecasting models. Extensive experiments were conducted on benchmark datasets with a large number of variables. Across various experimental settings, we achieved optimal results of nearly 90%.

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
