# OpenReview forum: "Boundary-Aware Periodicity-based Sparsification Strategy for Ultra-Long Time Series Forecasting"
_acmmm.org/ACMMM/2024/Conference — MM2024 Poster_

### Official Review · Reviewer_9DqV · 2024-05-23

**Rating:** 5
**Confidence:** 2

**Summary:**

First, the authors raise the urgent need in transportation, resource management, and weather forecasting for methods that can provide predictions over a sufficiently long time horizon. Compared with traditional time series forecasting, ultra-long time series forecasting needs to enhance the ability of the model to infer long time series while keeping the cost of inference within an acceptable range.To solve this problem, this paper proposes a boundary-aware periodic sparse strategy (BAP-UL) for very long time series prediction. This strategy is a general-purpose lightweight data sparsity framework reorganizes inputs and outputs into shorter subsequences for model prediction by capturing periodic features in time series. Combining the boundary characteristics of time series with the boundary sensing method improves the ability of the model to predict extreme peaks and irregular time series by adjusting the prediction results. The experimental results show that the BAP-UL model reaches the advanced technical level under some experimental conditions, and the author proves that the proposed cyclic sparse method has broad applicability.

**Strengths:**

1.The method presented in this paper achieves optimal results of nearly 90% experimentally on various datasets.
2.In this paper, many experiments prove that the method has a strong generalization, and even in other scenarios, it still achieved excellent results.
3.The method proposed in this paper is highly innovative，and addresses the need for methods to provide accurate forecasts over long time horizons in transportation, resource management, and weather forecasting. Compared with traditional methods, a new boundary-aware periodic sparse strategy (BAP-UL) is proposed, which can improve the capability of ultra-long time series prediction while keeping the cost acceptable.

**Limitations:**

Possible limitations: The method may rely on extracting periodic features of the input data and reassembling and predicting subsequences accordingly. However, not all time series data are obviously periodic, especially for aperiodic or irregular data, which may affect the effectiveness of this method.

**Suitability:**

3

---

### Official Review · Reviewer_RRfA · 2024-05-25

**Rating:** 4
**Confidence:** 4

**Summary:**

This paper is focused on the Ultra-Long Time Series Forecasting (ULTSF) task which consideres the time series forecasting problem with sufficently long forecasting horizon. Existing methods usually fail to accomplish this task with acceptable computational cost, so this paper proposed an Boundary Aware Periodicity-based sparsification strategy for this task. The periodicity-based sparsification strategy captures periodic features in time series and reorganizes inputs and outputs into shorter sub-sequences for model prediction. The boundary-aware method improves the model’s capability by adjusting the prediction results. Experimental results on three public benchmark datasets showed that the proposed method generally outperformed SOTA methods.

**Strengths:**

1. The time series data, as a special and important data modality other than texts/image/videos, is less discussed and researched by the MM community.
1. The problem studied in this paper, i.e., ULTSF, is meaningful in real world applications but less touched by existing research works.
2. It is novel to employ the FFT to sparsify the input data in terms of different periods/frequencies.
3. This paper is well written and easy to follow.

**Limitations:**

1. It is not clear why this paper claims the time series is bounded in nature, since the forecasting results should be an unbounded real values.
2. The experiment settings need to be improved and clearly stated. Most results of baseline methods are simply missing due to OOM, and none of the baselines finished all the three datasets in the three forecasting horizons, which makes it hard to comprehensively compare the performance of the methods. It is not clearly provided how much memory is allowed to conduct the experiments. Can the OOM issue be alleviated by using smaller batchsize, number of layers etc? How much memory is needed for the baselines to successfully run the experiment, and if they run, what will their performance be? If the model performance is bounded by the memory, what will the performance be in terms of different memory usage?

**Suitability:**

2

---

### Official Review · Reviewer_Co4M · 2024-05-26

**Rating:** 2
**Confidence:** 3

**Summary:**

This paper addresses the critical need for ultra-long time series forecasting over extended horizons to facilitate decision-making across various domains, proposing the Boundary-Aware Periodicity-based Sparsification Strategy for Ultra-Long Time Series Forecasting (BAP-UL). This paper introduces a novel data sparsification framework that captures periodic patterns inherent in time series data to reorganize input and output sequences into computationally tractable sub-sequences. Several experiments are conducted.

**Strengths:**

This paper studies an interesting setting that extra long time series forecasting.

**Limitations:**

1. The most important problem of this paper is the experimental results are less convincing. First, only three datasets are considered in this paper. Second, this paper compared with so many transformer-based forecasting methods, where most models would fail. The comparison is not meaningful. Third, the paper does not report univariate forecasting results.
2. The contribution is a lack of novelty. Previously, using the Fourier transform to extract periodic patterns has been tried in many works, such as TimesNet [1], DEPTS [2], etc. By the way, the 1d to 2d in the proposed method is very similar to TimesNet.

[1] Timesnet: Temporal 2d-variation modeling for general time series analysis. In ICLR.
[2] DEPTS: Deep expansion learning for periodic time series forecasting. In ICLR.

**Suitability:**

2

---

### Meta-Review · Area_Chair_kHuy · 2024-07-09

**Recommendation:** Accept (Poster)
**Confidence:** 5

**Metareview:**

This article has received 3 reviews. The 3 reviewers are confident. 2 reviews positive and propose the paper to be accepted. 1 review propose a rejection. After a deep analysis of this paper and all comments, I consider that the balance is positive and propose this paper for the conference

---

### Meta-Review · Senior_Area_Chairs · 2024-07-10

**Recommendation:** Accept (Poster)
**Confidence:** 4

**Metareview:**

This paper received mixed ratings initially. After rebuttal, two reviewers tend to accept the paper and one who gave negative rating did not submit the final rating. SAC and AC recommend acceptnce of the paper.